# Analysis on the Spatio-Temporal Evolution Characteristics of the Impact of China’s Digitalization Process on Green Total Factor Productivity

**DOI:** 10.3390/ijerph192214941

**Published:** 2022-11-13

**Authors:** Junwei Zhao, Yuxiang Zhang, Anhang Chen, Huiqin Zhang

**Affiliations:** College of Management Science, Chengdu University of Technology, Chengdu 610059, China

**Keywords:** green total factor productivity, digitalization, spatial heterogeneity, GTWR

## Abstract

Green production is an inevitable choice for China’s high-quality economic development. With the rise of the digital technology revolution, China’s digital transformation may play an integral and important role in increasing green total factor productivity (GTFP). Based on the panel data of 30 Chinese provinces from 2014–2020, the impact of digitization on GTFP was explored using the model of geographically and temporally weighted regression (GTWR), and the spatial and temporal distribution characteristics and development trends of such effects were further explored. The main findings are as follows: (1) China’s digitalization level and GTFP has significant spatial autocorrelation and similar spatial distribution characteristics. (2) Digitalization has a significant positive impact on GTFP, but this impact decreases yearly, and there are noticeable regional differences. Digitalization in the eastern and central regions has a more significant impact on GTFP than in the west. (3) The region where China’s digital development has extensively promoted GTFP has shifted from China’s southern coastal region to the northwest and northeast regions. (4) The time-series fluctuations of the regression coefficients of the digitization level in each province in China also show agglomeration characteristics. That is, the regression coefficients of neighboring provinces have similar time-series fluctuations.

## 1. Introduction

In recent years, China’s economy has been developing rapidly while facing the double pressure of environmental pollution and resource consumption. The World Energy Statistics Review 2020 states that China accounts for three-quarters of the growth in world energy consumption in 2019. In 2020, among 337 cities in China, 40.1% of the cities had air quality exceeding the standard, with a total of 1152 days of severe pollution. Among the 10,171 national groundwater-quality-monitoring stations, 86.4% of the monitoring stations contain Class IV or V water (China Ecological Environment Bulletin). China’s economic growth is gradually slowing down as environmental problems become more serious. Relevant data suggest that China’s GDP growth rate will be only 8.1% in 2021, 2.3 percentage points lower than in 2010 Against this backdrop, how to reconcile the relationship between the economy and the environment has become a topic of common concern in all sectors of society [1]. GTFP considers the environmental impact, meets the requirements of ecological civilization construction, and is a key indicator to measure the effectiveness of regional economic development. Therefore, scientific exploration of the key factors affecting GTFP will be beneficial to the healthy development of China’s society and economy.

Thanks to the opportunity of the digital technology revolution, China’s digitalization process has been accelerating. The China Academy of Information and Communication Technology points out that in 2020, China’s digital economy accounts for 38.6% of GDP, with an added value of CNY 39.2 trillion, of which the added value of industrial digitization is CNY 31.7 trillion, gradually becoming a new driving force for industrial structure upgrading and high-quality economic development. Especially under the COVID-19 epidemic, countries’ industrial chains began to contract, and international trade faced different degrees of decline. Digitalization can play its digital information advantage and better integrate and merge entity manufacturing enterprises through information changes, thus forming inter-enterprise data networks, reducing inter-enterprise information search costs, and improving the survival cycle of entity enterprises [2,3]. In this context, the Chinese government also attaches great importance to the digitalization process, clearly proposing to implement the new development concept, improve total factor productivity (TFP), guide the real economy in digital transformation, and build a digital China. This shows that an in-depth discussion of the impact of digitalization on GTFP is not only beneficial to the coordinated development of economy and society but also provides useful theoretical support for the implementation of the “Digital China” strategy and the leap forward in digital development.

The 19th Party Congress reported for the first time elevation of the regional coordinated development strategy to a regionally led development strategy, and regional economic ties are becoming increasingly close. Data and information, as new key factors of production in the digital economy, are different from the static and non-digital characteristics of previous information production factors and are storable and reproducible [4]. With the continuous development of digitalization, the barriers to the flow of information, data, technology, and talent between regions have been greatly reduced, which makes digitalization affect green total factor productivity with cross-regional spillover effects [5]. In addition, digitization may create new digital inequalities and “digital divides” and induce siphoning effects [6]. China has a large regional area, and the development of the eastern and western regions is significantly different. As the digital infrastructure in the western region is relatively inadequate, and the institutional environment, policy optimization, and other soft environments are relatively backward, digitalization may accelerate the flow of various factors to the eastern region, further widening the development gap and thus hindering the enhancement of GTFP. Therefore, considering the geographical location factor when exploring the relationship between digitalization and GTFP can make the research conclusions have a strong regional orientation and improve the accuracy of policy implementation.

This paper has the following contributions: (1) more comprehensive and objective evaluation results. This paper uses total energy consumption as energy input and wastewater discharge, SO_2_ discharge, and industrial solid wastes as non-desired outputs. GTFP of each province in the SBM-GML measure was used. We constructed a digital evaluation system from four aspects, namely digital infrastructure, digital inputs, digital economy, and digital applications, and calculated the level of digitalization using an improved entropy method. (2) Temporal and spatial factors are considered in the regression model. Given the potential for spatial spillover effects and interactions between neighboring regions for each variable, spatial measures were chosen for the analysis. It was also considered that both current and past conditions of the variables might have an impact on GTFP, so the time factor was considered in the regression analysis. (3) The relationship between digitization and GTFP was explored from the perspective of spatial and temporal variability, and on this basis, the spatial distribution characteristics and development trend of this influential relationship were further examined.

The other parts are organized as follows: The literature review is in the second part. The third part presents the related methods. The selection of variables and data sources are in the fourth part. The regression results and analysis are in the fifth part. Finally, research conclusions and policy implications are presented.

## 2. Literature Review

The traditional TFP is based on the impact of capital and labor on output and does not consider factors such as pollution emissions and ecological environment, thus failing to comprehensively and rationally evaluate the quality of economic development [7]. GTFP can categorize resources and environment into the analytical framework of productivity, which is more in line with the concept of green development in the new era [8], which has attracted widespread attention from scholars. Current relevant research mainly focuses on the measure of GTFP and its influencing factors.

### 2.1. Measurement of GTFP

GTFP measurement methods can be divided into parametric and nonparametric analysis methods. The former is mainly based on stochastic frontier analysis (SFA) [9]. The SFA model established on the specific production function can effectively measure the impact of random factors on production behavior. Cui et al. (2019) used stochastic frontier analysis (SFA) to evaluate GTFP growth trends across 36 industrial sectors in China [10]. Parametric method models are simple but require a prior determination of the functional form, and the use of the model is demanding, requiring accurate control and assumptions about the price information of the input and output variables. Data envelopment analysis is a commonly used nonparametric analysis method, which has obvious advantages in calculating multiple inputs and outputs. Pittman (1983) first applied DEA to consider undesirable outputs [11]. On this basis, Chung et al. (1997) and Fare et al. (2001) proposed an ML index that is more in line with the environmental concept [12,13]. Many scholars have used the method in subsequent studies. Li and Lin (2017) studied the impact of rationalization of China’s industrial structure on green productivity [14]. To solve the radial and angular problems of the distance function, based on Tone’s (2001) research, Fukuyama et al. (2009) constructed a non-radial, non-directional SBM directional distance function, which further reduced the measurement error [15,16]. In addition, the GML index proposed by Oh (2010) based on the ML index can avoid potentially infeasible linear programming and effectively solve nonlinear problems [17]. Song et al. (2018) applied the GML index method to the Chinese industrial sector, and the study showed that the GTFP of the industrial sector decreased after considering environmental and energy constraints [18].

It can be seen that both the GML index and the SBM directional distance function make up for the shortcomings of the previous methods, but there are still some problems when used alone. The former cannot reduce deviations caused by radial and orientation problems. The latter cannot effectively handle the inconsistency of the production front at each production unit stage, thus affecting the comparison of results during the period [19]. Given the limitations of both, some scholars proposed the GML index based on the SBM direction distance function. This method has the advantages of both, so it is favored by scholars. Fang et al. (2021) found an upward trend in China’s agricultural GTFP by province from 2002 to 2015 [20].

From the above analysis, we can see that the GML index method based on SBM directional distance function has obvious advantages in calculating GTFP, but it should be noted that scholars in different fields pay different attention to the indicators. China has a vast territory and significant differences in regional development, and the selection of indicators should be as fair as possible not only to reflect the state of the ecological environment of each province but also to reasonably evaluate the quality of economic development. Therefore, this paper constructs the corresponding index system from the four aspects of economy, population, energy consumption and pollution emission so as to provide a reasonable basis for this study.

### 2.2. Influencing Factors of GTFP

Concerning the factors influencing GTFP, most scholars focused on exploring the impact of various factors in the agricultural [20,21], industrial [22,23], and service sectors [24] on GTFP. In addition, Wang et al. (2021) showed that the carbon emission trading system has a more obvious role in promoting GTFP in regions with a high degree of marketization and a relatively low proportion of coal consumption [25]. Guo et al. (2021) examined the relationship between low-carbon pilot policies and GTFP and found that low-carbon pilot policies can promote GTFP through industrial structure optimization [26]. Lee et al. (2022) pointed out that both environmental regulation and innovation ability can promote GTFP, but there is heterogeneity in the impact of innovation ability on GTFP [27]. Based on the panel data of the first batch of smart city construction pilots in China, Jiang et al. (2021) empirically investigated the impact mechanism of smart city construction on GTFP [28].

Some scholars believe that digital technology can reduce resource consumption and environmental pollution and improve resource utilization efficiency while maintaining stable economic development [29,30]. Habanik et al. (2019) believed that the use of digital technology in all areas of society contributes to sustainable economic and social development [31]. Research by Pan et al. (2022) showed that the digital economy is an innovation driver for TFP development [32]. Li et al. (2020) used a fixed-effect model to empirically investigate the impact of Internet development on GTFP, and the results showed that the Internet can promote GTFP by integrating resources [33].

However, research and debate about the IT productivity paradox have persisted since Solow proposed it [34,35]. Groves et al. (2013) showed that big data and the Internet of Things are still in their early stages, and their practical effects are still to be studied [36]. Acemoglu and Restrepo (2018) performed a study suggesting that excessive informatization may lead to wasted resources and misallocation of labor, which indirectly inhibits TFP growth [37]. By analyzing the digitization process and economic growth, some other scholars argue that the role of digitization varies widely across regions. Ye et al. (2020) believed that digitalization can provide strong support for economic development in backward regions [38]. However, Liu et al. (2022) found that the digital economy has a more obvious role in promoting GTFP in the eastern and central regions of China [39]. Pan et al. (2022) also had similar research conclusions [32]. They believed that the digital economy in the western regions has a significantly weaker role in promoting TFP than in the east.

Generally speaking, the above scholars’ research provides theoretical reference and design ideas for the discussion of the relationship between digitization and GTFP, but it should also be pointed out that the above research still has shortcomings; on the one hand is a lack of consideration of time and space factors. As the impact of digitization on GTFP depends not only on current information but also on previous information, time effect needs to be incorporated into the regression model. In addition, the development of a region can not only affect the surrounding areas but also be affected by the surrounding areas [40,41,42]. Considering that the influence of digitization on GTFP may have spatial spillover effect and the interaction between adjacent regions, it is necessary to consider both time and space factors when discussing the relationship between digitization and GTFP. On the other hand, the existing researches pay more attention to the impact of digitization on GTFP, but there is no in-depth research on the spatio-temporal distribution and evolution trend of the impact of digitization on GTFP. In view of this, based on the relevant data of 30 provinces in China from 2014 to 2020, this paper uses the GTWR model to explore the impact of digitization on GTFP from the perspective of spatio-temporal differences. On this basis, the spatial distribution and temporal fluctuation of this influence are further investigated in order to provide more accurate evidence for decision making and policy making.

## 3. Method

### 3.1. Spatial Correlation Analysis Method

Spatial correlation analysis is used to test whether a phenomenon has agglomeration in space. Commonly used methods are Moran’s I index and Geary’s C index. We used the Moran I index to test whether neighboring regions within a region are similar (positive spatial correlation), different (negative spatial correlation), or independent (random distribution). It is calculated as follows:(1)$$I=\frac{N\sum_{i=1}^{n}\sum_{j=1}^{n}\omega_{ij}(x_{i}-\bar{x})(x_{j}-\bar{x})}{(\sum_{i=1}^{n}\sum_{j=1}^{n}\omega_{ij})\sum_{i=1}^{n}{\left(x_{i}-\bar{x}\right)}^{2}}$$
where N is the number of spatial units in the study area, $\omega_{ij}$ is the spatial weight, $\bar{x}$ is the mean of the attributes, and $x_{i}$ and $x_{j}$ are the attributes of space i and space j, respectively.

### 3.2. GTWR Model

The traditional linear regression model only estimates the independent variable parameters as a whole and does not take into account the relationship between the geographical units, which may lead to some deviation in the analysis results. With the gradual development of spatial econometric analysis technology, scholars pay more and more attention to the influence of spatial factors. The geographically weighted regression model (GWR) effectively overcomes the spatial heterogeneity among geographical units and can draw differentiated research conclusions for different regions. However, its deficiency is that it can only carry on the regression analysis to the cross-section data. when there are too many parameters to be estimated, it will greatly lose the accuracy of parameter estimation. Therefore, Huang et al. (2010) proposed the model of GTWR, which effectively makes up for the weaknesses of the GWR model [43].

The GTWR model embeds the spatio-temporal characteristics of the research data, estimates the parameters of different time points in the process of geographical location change, and carries out local regression to each observed spatial unit, which can not only better reflect the spatio-temporal differences of the driving factors but also make the parameter estimation and statistical test results of the model more significant. Using it to analyze the impact of China’s digitization process on GTFP, we can better clarify the action mechanism and intensity of provincial digitization on GTFP and then accurately reveal the differences in the degree of influence. In addition, compared with the general regression model, the GTWR model can also use software such as ArcGIS to visualize the regression parameters of each sample point in space, which makes the performance of the model results more intuitive. The specific formula is as follows:(2)$$y_{i}=\beta_{o}(u_{i},\nu_{i},t_{i})+\sum_{k=1}^{d}\beta_{k}(u_{i},\nu_{i},t_{i})x_{ik}+\varepsilon_{i},i=1,2,\cdots ,n$$
where $y_{i}$ is the dependent variable, $x_{ik}$ represents the independent variable, $\left(u_{i},\nu_{i},t_{i}\right)$ is space-time information of province i, $\beta_{o}$ is the regression constant, $\beta_{k}$ denotes the regression parameter, and $\varepsilon_{i}$ is the residual.

## 4. Variables and Data

### 4.1. Dependent Variable

To enhance the rationality and accuracy of the study as much as possible, after comparing the existing measurement methods, the GML index method based on SBM directional distance function was selected to calculate GTFP [17,19]. The specific indicators are shown in Table 1. The number of people employed at the end of the year represents an indicator of labor input. The capital stock represents the capital investment index. Since there are no official statistical data on the capital stock in each province in China, this study adopts the perpetual inventory method for estimation. The base year uses the 2000 capital stock data measured by Zhang et al. (2004) [44]. Referring to the existing research [45,46,47], this paper chooses total energy consumption to represent energy input. For the measurement of undesired outputs, studies have shown that simply using one pollutant emission indicator may lead to biased results, which may overestimate or underestimate pollution levels [48]. Based on the availability of data, taking into account the actual situation of China’s pollution emissions, this paper selects total wastewater discharge, SO_2_, and solid waste to measure the undesired output.

The GTFP calculated by the GML index method is dynamic. Referring to Liu et al. (2019), if the GTFP of the first year is set to 1, the GTFP of the *t* + 1 year is as follows:(3)$$GTFP_{t+1}=GML_{t}^{t+1}*GTFP_{t}$$

### 4.2. Main Independent Variable

Regarding the measurement of digitization level in China, there is a lack of uniform standards. A single indicator can only reflect part of the process and characteristics of digitalization, and it is difficult to reflect the complex and comprehensive digitalization level. This paper believes that digitization itself has rich meanings. To evaluate its development level, consider not only economic value-added and social impact but also the digital infrastructure that affects its level of development. Therefore, it is more appropriate to use multiple indicators to measure the level of digital development. Based on the social informatization indicators published by the International Telecommunication Union, the China Economic Development Index published by the Tencent Research Institute, and the research of scholars such as Li et al. (2021) and Liu et al. (2022) [39,48], the level of digitization (DIG) is measured in four dimensions: digital foundation, digital input, digital economy, and digital application. The specific index system is shown in Table 2.

We chose the entropy method for weighting, which can avoid the bias caused by human factors. Since it is aimed at panel data, related improvements are made on the basis of the entropy method, and a time factor is introduced. If there are *θ* years, n indicators, and m provinces, $x_{\mathrm{tij}}$ denotes index j of province i in year *t*, where t=1,2,⋯,θ; j=1,2,⋯,n; i=1,2,⋯,m.

1.Normalization of indicators:


(4)
$$\mathrm{Normalization}\;\mathrm{of}\;\mathrm{positive}\;\mathrm{indicators}\text{: }x_{tij}^{\prime }=\frac{x_{tij}}{\max\left\{x_{tij}\right\}}$$



(5)
$$\mathrm{Normalization}\;\mathrm{of}\;\mathrm{negative}\;\mathrm{indicators}\text{: }x_{tij}^{\prime }=\frac{x_{tij}}{\min\left\{x_{tij}\right\}}$$


2.Calculate the weight of $p_{tij}$:


(6)
$$P_{tij}=\frac{x_{tij}^{\prime }}{\sum_{t=1}^{\theta }\sum_{i=1}^{m}x_{tij}^{\prime }}$$


3.Calculate the entropy value of each index $e_{j}$:


(7)
k=ln(θm)



(8)
$$e_{j}=-k\sum_{i=1}^{m}P_{tij}\ln(P_{tij})$$


4.Calculate the weight of $w_{j}$:


(9)
$$w_{j}=\frac{1-e_{j}}{\sum_{j=1}^{n}\left(1-e_{j}\right)}$$


5.Calculate the comprehensive score of the digitalization level of each province:


(10)
$$S_{ti}=\sum_{j=1}^{m}w_{j}x_{tij}^{\prime }$$


### 4.3. Control Variables

To accurately analyze the impact of digitization on GTFP and reduce the bias caused by missing variables, we selected the following five control variables based on existing studies [26,27,28]: (1) upgrading of industrial structure (STR), expressed as the share of the value added of tertiary industries in regional GDP; (2) labor level (LAB), in terms of year-end resident population; (3) the degree of government intervention (GOV), measured by the share of government fiscal spending in regional GDP; (4) the level of opening to the outside world (OPEN), measured by the total value of imports and exports as a percentage of regional GDP; and (5) economic development level (GDP), measured by the gross domestic product. To eliminate heteroskedasticity, some variables are logarithmically treated.

### 4.4. Data

The data mainly come from the China Statistical Yearbook, the China Environmental Statistical Yearbook, the China Energy Statistical Yearbook, the China Urban Construction Statistical Yearbook, the China Population Statistical Yearbook, and the National Bureau of Statistics. Data from Tibet, Hong Kong, Macau, and Taiwan were excluded, as they were severely missing. Finally, data from 2014–2020 for 30 provinces across the country were collected as samples, and the data of related variables are shown in Table 3.

### 4.5. Multiple Collinearity Test

This paper uses Stata16.0 to perform a multicollinearity test on the selected variables; the results are shown in Table 4. The multicollinearity test indicates that the VIF value of each variable is less than 5, the maximum value is 4.65, the minimum value is 1.79, and the mean value is 3.13, indicating that there is no multicollinearity between variables.

## 5. Empirical Results and Discussion

### 5.1. OLS Model

Due to the weak diagnostic results of GTWR, regression analysis using the OLS model was required before analysis to ensure the validity of the results. As shown in Table 5, all variables are significant and can be included in the GTWR model for regression analysis. Among them, digitization has the most significant positive impact on GTFP, indicating that digitization is conducive to improving GTFP, which is also in line with the research conclusions of Liu et al. (2022) and Li et al. (2020) [33,39].

### 5.2. Spatial Difference Distribution

The distribution of GTFP in China’s provinces from 2014 to 2020 is shown in Figure 1. This paper notices that the distribution of GTFP has significant spatial differences. As shown in Figure 1, Beijing’s GTFP in 2020 (2.34) is the highest in the country, followed by Shanghai (1.97), with Tianjin (1.85) in third place. Coastal regions such as Guangdong (1.77), Jiangsu (1.75), Shandong (1.67), and Zhejiang (1.64) also maintain high levels. Judging from the average annual growth rate from 2014 to 2020, Beijing, Shanghai, and Tianjin still occupy the top three, with average yearly growth rates of 10.66%, 10.03%, and 9.76%, respectively. Next are Shandong (7.81%), Hebei (7.54%), Jiangsu (7.39%), and other eastern coastal areas. In contrast, Guizhou (0.69%), Heilongjiang (0.38%), Gansu (−0.36%), and Xinjiang (−0.92%) had lower annual growth rates, among which the GTFP performance of Gansu and Xinjiang was average negative growth. In general, the GTFP of China’s provinces shows an upward trend year by year, but the gaps between provinces are significant, showing a state of aggregation in space: the southeast coastal area has a higher level and rapid development, followed by the central area, the western area, and the northeast area, which have a lower level, and some areas show negative growth.

Figure 2 shows the digitization levels of China’s provinces from 2014 to 2020. It shows China’s digitization level and GTFP have similar spatial distribution characteristics. As shown in Figure 2, the regions with a higher level of digitalization in 2020 are Beijing (0.64), Guangdong (0.45), Shanghai (0.37), Zhejiang (0.34), Jiangsu (0.31), and other places. Except for Beijing, all of them are on the southeast coast. In contrast, the digitization of Ningxia (0.091), Xinjiang (0.090), Gansu (0.087), and Heilongjiang (0.081) is at a lower level. Looking at the average annual growth rate from 2014 to 2020, Qinghai has the highest average annual growth rate (34.18%), followed by Guizhou (30.04%) and Ningxia (27.61%). The level of digitalization in Shanghai, Beijing, Jiangsu, Tianjin, and other places is still in the stage of rapid increase, but the average annual growth rate has slowed down, reaching 16.22%, 15.23%, 12.81%, and 10.98%, respectively. From the perspective of the spatial pattern, although the digitalization of the western region is developing rapidly, there is still a big gap with the eastern region. The digitization level of Chinese provinces also shows a high level in the east and a low level in the west.

### 5.3. Spatial Correlation Characteristics

In order to analyze the spatial correlation between digitization level and GTFP in each province of China, this paper is based on the GeoDa platform, using Moran’s I index method, and under the condition that the positive statistic z exceeds the critical value of 1.96 at the 5% significance level, the global Moran’s I index of China’s GTFP and digital development level are calculated. The results are shown in Figure 3.

The spatial autocorrelation analysis of GTFP and digitization level shows that the z value is more significant than 1.96, with apparent spatial autocorrelation. The Moran’s I value for the level of digitization has a maximum value of 0.278 and a minimum value of 0.170, showing an overall decreasing trend, indicating that the positive spatial correlation of the level of digitization development weakens over time. Since 2014, the Moran’s I value of GTFP has fluctuated, but it has generally remained above 0.3, with strong spatial autocorrelation.

### 5.4. Significance Test

The GTWR model was run through ArcGIS software, and an adjusted R2 of 94.46% was obtained, which is much higher than the 72.77% of the OLS model. It shows that the fitting effect of GTWR is better. To verify the reliability of the GTWR regression results, the Moran index method was used to test the residuals of the regression results. If there is no spatial correlation between the residuals, the regression results of GTWR are reliable [42]. As Table 6 shows, since all residuals were tested for significance by z-values, and the *p*-values were all greater than 0.05, the residuals were randomly distributed, and the regression results were reliable.

### 5.5. GTWR Regression Results and Analysis

The GTWR can estimate the local effects of each variable in its spatial and temporal evolution, and its parameter estimates vary with the spatial and temporal evolution. To explore the regional heterogeneity of the impact of various factors on GTFP, we divided China into eastern, central, and western regions. The GTWR regression results for each region are shown in Figure 4. The *x*-axis illustrates the influencing factors, where 0–1 illustrates DIG, 1–2 illustrates OPEN, 2–3 illustrates STR, 3-4 illustrates LAB, 4-5 illustrates GOV, 6–7 illustrates GDP, and each factor is divided into four sections presented according to east, central, west, and national. The years are expressed on the *y*-axis, and the GTWR regression results are expressed on the *z*-axis. Figure 4 shows that DIG has the most significant impact on GTFP, followed by STR and OPEN, while LAB, GDP, and GOV have less impact. In addition, the impact of the same factor on GTFP varies widely across regions, and the degree of impact of the same factor on GTFP in the same region varies over time. This suggests that the level of digital development and other factors affect GTFP differently in both temporal and spatial dimensions. Therefore, it is necessary to explore the influence of various factors on GTFP from time and space.

As shown in Figure 4, from the perspective of the spatial dimension (*x*-*z*), DIG, OPEN, STR, and LAB have more significant spatial characteristics, so these four factors were analyzed from the spatial dimension.

(1) DIG positively affects GTFP, showing the characteristics of “strong in the eastern and central regions, and weak in the western regions”. Digitalization can effectively combine emerging technologies such as the Internet of Things and cloud computing, significantly reducing the excessive consumption of energy and resources in traditional industrial production, thereby improving the utilization efficiency of factors [49]. On the other hand, digitalization has contributed to green technology innovation [50]. Compared with the west, the digital foundation, digital economy and digital application level of the eastern region are higher, which can more effectively promote the deep integration of information technology and traditional industries, thereby accelerating the improvement of GTFP.

(2) OPEN has a positive effect on GTFP, which is consistent with the conclusion of Ding et al. (2022) [51]. Opening to the outside world can effectively introduce foreign capital, strengthen economic ties inside and outside the region, and bring advanced production technology, management experience, and other resources to the region’s product development and stimulate the growth of GTFP. The positive impact of the level of external opening on the western region is the greatest, probably because there is still a gap between the western region and the central and eastern regions in terms of production equipment and technology, and the introduction of foreign investment can significantly improve the production equipment and technology level. In addition, the eastern and central regions, with their higher economic status and significant globalization, there is reliance on the introduction of foreign investment to improve technology and strengthen management to a limited extent, so the degree of influence is weaker than that of the western region.

(3) STR positively impacts the GTFP, showing the characteristics of “strong central, followed by western, and weak eastern”. As we all know, the rise of the low-pollution tertiary industry will be more conducive to promoting GTFP. As the proportion of the tertiary industry increases, energy consumption and pollution emissions are gradually reduced, and resource allocation is optimized, thereby improving GTFP. Currently, high-polluting industries in eastern China are progressively shifting to the central region, and industries with high energy consumption are concentrated in the central region. Therefore, optimizing the industrial structure in the central region will be more conducive to the promotion of GTFP.

(4) LAB has a positive impact at national and regional levels, showing a “strong western and weak eastern” character. The eastern region has a significantly higher level of economic development than other regions. New industries and new business models are emerging at an accelerated pace, attracting a large amount of labor and capital, but due to the heterogeneity of the labor force factors exist, this movement is not all highly qualified labor migration [52]; if the proportion of low-skilled labor movement is too high, it is not conducive to the GTFP increase, and the concentration of population will lead to the concentration of industrial industries, which will lead to a significant increase in resource consumption and various pollution emissions, which will, in turn, hinder the GTFP.

From the time dimension (*yz*) perspective, DIG, OPEN, STR, GOV, and GDP have more significant time characteristics. Therefore, these five factors were analyzed from the time dimension.

(1) The positive impact of DIG on GTFP gradually diminished over time. Currently, China’s information infrastructure is relatively complete, and the effect of digitalization on GTFP is reflected in the rapid development of consumer Internet and e-commerce. However, digital industrialization and industrial digitization are immature [53]. Compared with digital infrastructure, the digital industry is the driving force behind the continuous improvement of GTFP [54]. Therefore, for regions with better digital infrastructure to continue to promote GTFP, it is necessary to focus on the development of digital industries.

(2) The positive promoting effect of OPEN on GTFP has been continuously enhanced over time. Opening to the outside world can expand production scale and improve resource allocation efficiency, thereby promoting economic development [55]. In the early days, fewer cities opened up to the outside world, mostly concentrated in the southeastern coastal region; as internationalization accelerated, more and more cities began to join in international trade, so the positive influence gradually increased.

(3) The positive promoting effect of STR on GTFP is shown as rising first and then falling. At the beginning of the study period, China was mainly dominated by primary and secondary industries, and upgrading the industrial structure could significantly increase GTFP. With the rise of tertiary industries, China’s industrial structure gradually rationalized, and its positive impact on GTFP began to diminish.

(4) The influence degree of GOV on GTFP showed that it first increased and then decreased and changed from positive influence to negative influence. Due to the many unreasonable aspects of China’s development in the early stage, the government’s participation in appropriate regulation is more conducive to improving GTFP. With the gradual rationalization of economic growth, excessive government intervention may increase the production cost of enterprises, which will lead to the reduction of R&D investment by enterprises, which is not conducive to the promotion of GTFP [56].

(5) The positive effect of GDP on GTFP is becoming more and more prominent. China’s early economic development was relatively crude, and problems such as ecological damage and environmental pollution came to the fore [57]. With the increase in technology in all areas, the importance of the environment has become more important, and green development has been promoted, no longer sacrificing the environment for economic growth. Therefore, the impact of economic development on GTFP has changed from negative to positive.

### 5.6. The Impact of Digitization on GTFP from the Perspective of Spatiotemporal Heterogeneity

#### 5.6.1. Spatial Differences in Coefficients

To explore the spatial differences in the impact of digitization on GTFP, the regression coefficients for digitalization in 2014 and 2020 are presented visually in this paper, as shown in Figure 5. Digitization has a significant impact on GTFP. The parameter differences between adjacent provinces are slight. At the beginning of the research period (2014), the places where digitization had a more significant impact on GTFP were mainly concentrated in the southeastern coastal areas. At the end of the study period (2020), it was transferred to northern China.

The regression parameters for digitization levels in Guangdong, Fujian, and Hainan show a decreasing trend but still a positive contribution. As China’s first economic province, Guangdong Province has a sound economic system and a good foundation for building a modern economic system. The electronic information industry, with an output value of over USD 1 trillion as early as 2005, has become the province’s first industry, so the province’s digital infrastructure is well-established. At the same time, Guangdong Province is also the most populous province in China. With the advantages of excellent data infrastructure and human capital accumulation, it is easier to guide the industry from value remodeling to value creation, eliminate dependence on traditional factor paths, and accelerate the growth of GTFP. This is also in line with Pan et al. (2022) and Liu et al. (2022) that digital transformation in regions with better foundations will be more conducive to improving GTFP [32,38]. In Guizhou, Fujian, and other provinces adjacent to Guangdong Province, although the level of digital development is relatively backward, due to the continuous acceleration of the digital process, the barriers to the flow of information, data, technology, and talents between regions have been significantly reduced. Regions with a high level of digital development can also stimulate neighboring regions through spillover and demonstration effects, so these places rely on the advantages of Guangdong, especially the electronic information industry, to vigorously promote the construction of extensive comprehensive data pilot areas and develop the big data industry so that the GTFP in the province has also been rapidly improved. However, if digital development only depends on increasing digital infrastructure carriers, green total factor productivity can only be improved in the short term. In the long run, this effect will decay over time [54]. Therefore, if these cities are to be able to exploit the green value of digital development in the long term, they need to speed up the process of digitization and digital industrialization of industries, cultivate new industries and new business models, and provide a lasting source of power for the green development of the cities.

The regression parameters of the digital development level in Qinghai, Gansu, Inner Mongolia, Heilongjiang, Jilin, and other regions showed a significant upward trend. These provinces are located in the northwest and northeast regions of China, with backward economic development levels, serious brain drain, and a late start and slow development of digital development. With the acceleration of China’s digitalization process, and influenced by the Western Development Strategy and the Northeast Revitalization Strategy, policy guarantees have created favorable conditions for developing the big data industry in these places. In addition, the scale effect of the industry should also not be ignored. These provinces are located inland, and their geographical location and level of development make their layout of the digital industry lag far behind that of developed coastal areas, but their industrial planning can take advantage of the latecomer advantage and draw on existing experience to develop a more efficient and realistic development path and management system, thus maintaining a high output efficiency. Although Xinjiang is also on the rise, it still has a negative impact. Xinjiang is located in the northwest of China. It is a late starter in terms of digital development, which is further limited by poor infrastructure and lack of human resources as well as by the low level of digitalization in neighboring provinces, which makes digital development difficult and costly and therefore harms GTFP.

#### 5.6.2. Timing Fluctuations in Coefficients

To visually show the time-series fluctuation of the impact of digitization on GTFP, the time-series fluctuation is drawn according to the regression analysis results. The *x*-axis represents different years (2014–2020), and the *y*-axis represents the size of the coefficient, as shown in Figure 6.

As shown in Figure 6, the time-series fluctuations of coefficients can be roughly divided into four categories: “falling-rising type”, “rising type”, “falling type”, and “fluctuation type”. Among them, the falling-rising areas are mainly concentrated in the northern part of China, including Beijing, Tianjin, Inner Mongolia, Liaoning, Shanxi, and other regions. The rising type includes Heilongjiang and Jilin, located in the northeastern part of China. The falling type is mainly located in the central and southeastern coastal areas of China, including Zhejiang, Jiangsu, Fujian, Guangdong, Hubei, Hunan, and other regions. The rest of the provinces are fluctuation type, mainly located in the western part of China.

To see the distribution of the four time-series fluctuations more intuitively, we used ArcGIS to visualize them. It can be seen from Figure 7 that the time-series fluctuations of the regression coefficients of the digitization level in each province in China also show agglomeration characteristics. The regression coefficients of adjacent regions show the same time-series fluctuations.

### 5.7. Comparison of Provincial Regression Results

To facilitate a comparison of regression coefficients between provinces at an average level, the paper divides the 30 regions into four categories (divided by different colors) based on the average annual growth rate (*x*-axis) and the average value (*y*-axis) of the regression parameters for each province (Figure 8). The majority of Chinese provinces have positive values for the digitization level regression parameters, indicating that digital development contributes to GTFP.

Judging from the size of the regression coefficient, when the points are located in areas I and II, these provinces’ regression coefficient is higher than the average level (1.46). Below-average points are distributed in areas III and IV. It can be seen from the figure that the regression coefficients of most provinces are close to the average level. Among them, the regression coefficient of Xinjiang is relatively small, significantly lower than the average level. The regression coefficients in Guangdong, Fujian, and other places are higher than the average level, and digitization in these places has a more significant impact on GTFP.

From the point of view of the average annual growth rate, when the points are located in areas I and III, it indicates that the annual average growth rate of the regression coefficient of these provinces is lower than the average level (−6.2%), mainly in Guangdong, Shanghai, Zhejiang, and so on. The points in areas II and IV indicate that the annual average growth rate of the regression coefficients in these provinces is higher than the average level. There are mainly Shaanxi, Qinghai, Beijing, and so on. These regions have contributed to the growth of the national regression coefficient.

According to the above analysis, we should focus on areas with smaller average regression coefficients and annual growth rates, such as Xinjiang, Hainan, Guizhou, Guangxi, Fujian, etc.

## 6. Discussion and Policy Implications

With the rise of the information technology revolution, digital development has become an indispensable important factor affecting global economic growth and environmental quality improvement. Under the double constraints of resources and environment, it is urgent to guide the transformation of China’s economic development model to intensive and improve GTFP. Therefore, it is of great practical significance to explore the impact of digital development of various provinces in China on GTFP. However, the existing studies pay more attention to the impact of digitization on GTFP, and there is no in-depth discussion on the spatio-temporal distribution and evolution trend of the impact of digitization on GTFP. Therefore, based on the relevant data of 30 provinces in China, this paper empirically explores the influence of the digitization process of Chinese provinces on GTFP from 2014 to 2020 by using the GTWR model, accurately reveals the development difference of the influence degree, and further examines the spatial distribution and time series fluctuation of this influence. The main conclusions are as follows:

(1) The digitization level of Chinese provinces has a similar spatial distribution to GTFP, showing a gradually decreasing trend from east to west. In addition, the spatial correlation of GTFP fluctuates wildly, and the overall level is higher. The spatial correlation of digitization levels decreases year by year, indicating that the degree of digitization interaction between adjacent regions is weakening.

(2) The regression results of both the OLS and GTWR models indicated that digitization had a significant positive effect on GTFP. However, judging from the changes in the regression coefficients of GTWR, this positive effect is decreasing year by year and shows noticeable regional differences. That is, the impact of digitization on GTFP is greater in the eastern and central regions than in the western region.

(3) From different perspectives, there are spatial and temporal differences in the impact of digital development on GTFP.

From a spatial point of view, at the beginning of the study period (2014), the southern coastal areas such as Guangdong, Fujian, and Hainan had a more substantial promotion effect, while Gansu, Qinghai, Heilongjiang, and Jilin had a weaker promotion effect. At the end of the research period (2020), the northwest and northeast regions such as Gansu, Qinghai, Inner Mongolia, Heilongjiang, and Jilin had a strong promotion effect, while Guangdong, Fujian, Hunan, and other places had a weak promotion effect. This suggests that the region where GTFP has been extensively promoted by China’s digital development has shifted from China’s southern coastal region to the northwest and northeast regions.

From the perspective of time, the time-series fluctuations of the digitalization-level coefficients in northern regions such as Beijing, Tianjin, Hebei, and Shandong show a decrease first and then an increase. Shanghai, Jiangsu, Zhejiang, Hubei, Hunan, and other central and eastern regions show a decline. Heilongjiang and Jilin show an upward trend. The time-series fluctuations of the coefficients of other provinces are unstable, sometimes rising and sometimes falling. This indicates that the time-series fluctuations of the regression coefficients of the digitization level in each region in China also show agglomeration characteristics. That is, the regression coefficients of adjacent provinces have similar time-series fluctuations.

(4) Comparing the average annual growth rates and regression coefficient averages across provinces shows that Xinjiang has a smaller regression coefficient, lowering the national average. Although the impact of digital development in Guangdong, Fujian, and Hainan on GTFP has increased the national average, it has shown negative growth. Both Heilongjiang and Jilin increased the average annual growth rate of the national regression coefficient.

China has a vast territory and significant differences in regional development, so various regions should adopt development strategies in the light of the actual situation and in accordance with local conditions. Based on the conclusions of this paper, the following suggestions are put forward:

(1) It is recommended to adopt a differentiated digital development strategy and unswervingly build digital China [39]. For the eastern regions with talent and capital endowment advantages (such as Beijing, Shanghai, etc.), it is important to continue to give full play to the advantages of resources, maintain innovation vitality, and constantly transform the achievements of digital innovation into productive forces and transfer to the central and western regions [32]. For the central region (such as Hubei and Jiangxi), we should combine the advantages of advanced manufacturing core areas, actively promote industrial digital transformation, and make full use of digital technology to realize the efficient flow of various factors of production and improve the efficiency of resource allocation. The western regions (such as Gansu and Shaanxi) should continue to give full play to their institutional and late-developing advantages [6], strengthen the construction of digital infrastructure, and improve their ability to undertake the transfer of innovative achievements in the eastern and central regions.

(2) The development of digitalization depends more on the market driver based on application innovation [58]. The lack of technological innovation caused by the lack of core technology and the shortage of high-skilled personnel has become an important reason why digitalization cannot continuously and effectively improve GTFP. Therefore, for the areas where the digitization level is high, but the continuous promotion is not strong (such as Jiangsu, Guangdong, etc.), we should increase the R&D intensity and capital investment of the core technology to realize the key core technological innovation in the digital field. For the areas with low levels of digitization and a large population size and mobility (such as Sichuan, Henan, etc.), we should continue to consolidate the digital foundation and devote ourselves to the cultivation of digital talents. With the development of digitalization, some repetitive jobs will be gradually replaced by machines, resulting in higher and higher requirements for the quality of talents [59]. Therefore, these regions can meet their own talent needs while also providing more quality talent to advanced regions.

(3) Based on the agglomeration characteristics of time-series fluctuations, we can know that the digital development of falling-rising areas (such as Inner Mongolia, Hebei) and rising areas (Heilongjiang, Jilin) can better promote GTFP, so we should continue to increase digital investment. Declining areas (such as Shandong, Zhejiang, etc.) have good digital infrastructure and generally have a high level of digitalization. In order to continuously improve GTFP, we need to speed up the process of digital industrialization and industrial digitization and cultivate new industries and new business types. The digital development of fluctuating areas (such as Qinghai, Ningxia, etc.) starts late, and the brain drain is serious, so we should fully learn from the existing experience, improve the welfare of talents, and carry out digital transformation in accordance with local conditions.

(4) We must give full play to the role of digital factor allocation, reduce the flow barriers of production factors [4], and strengthen regional linkage. For example, Guangdong, Hainan, and Fujian should break the restrictions on administrative divisions and strengthen cooperation closely. Advanced areas (such as Guangdong Province) should steadily promote digital development and avoid blind expansion and disorderly development. For the surrounding areas (such as Guangxi, Hainan, etc.), we should give full play to the enabling effect of digitization on traditional industries, actively undertake technology spillover from advanced areas, and constantly explore new momentum of economic growth while avoiding the loss of elements as far as possible.

## 7. Conclusions

This paper explores the impact of China’s digitalization process on green total factor productivity from the perspective of spatial and temporal heterogeneity, which enriches the related research, but there are still some limitations. First of all, the measurement of China’s digital development level is still controversial, and there is no unified standard; yet, we will further explore a more current and authoritative standard. Second, this paper uses provincial data and focuses on the impact of the digitalization process on green total factor productivity at the provincial level, while a more detailed exploration of the relationship between the two at the city level will be the focus of our future research. Third, we analyzed the impact of China’s digitalization process on GTFP using the GTWR model, but it only reflects the characteristics of the data and the impact relationship in 2014–2020, and it is also worthwhile to further explore whether new characteristics are present today or what kind of changes have occurred.

## Figures and Tables

**Figure 1 ijerph-19-14941-f001:**
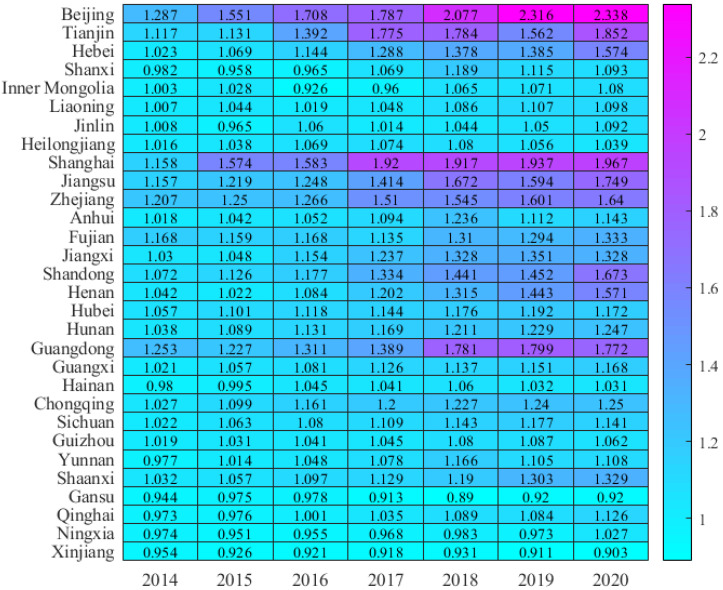
GTFP.

**Figure 2 ijerph-19-14941-f002:**
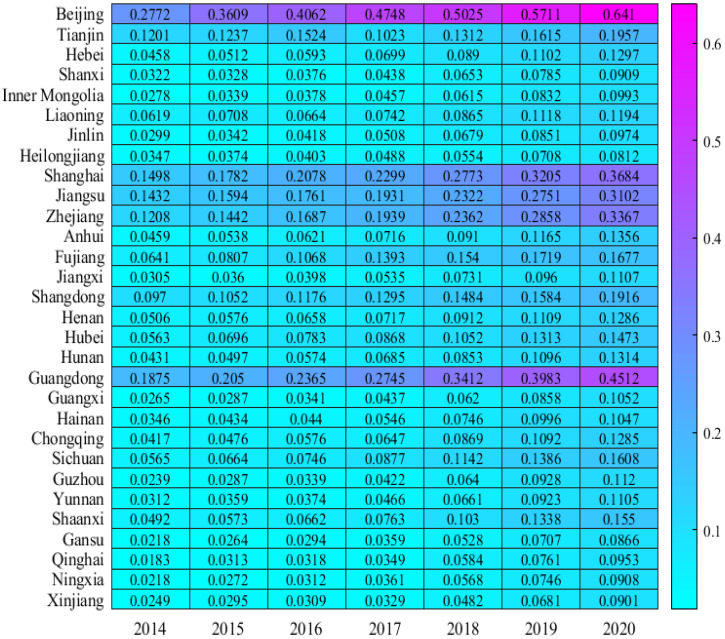
Digitization.

**Figure 3 ijerph-19-14941-f003:**
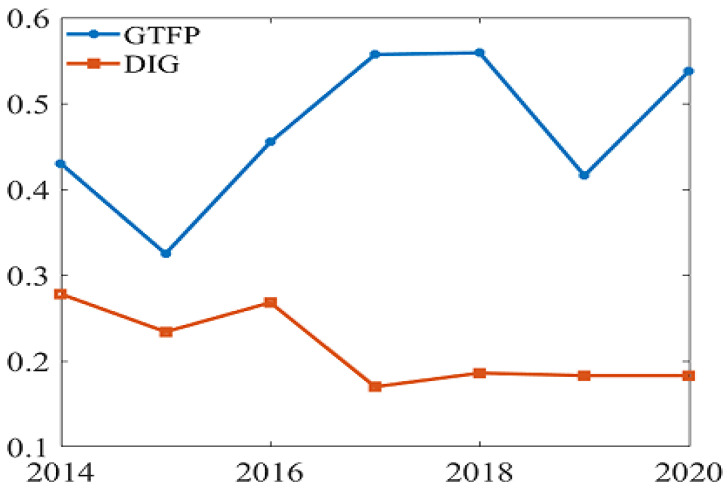
Moran’s I during 2014–2020.

**Figure 4 ijerph-19-14941-f004:**
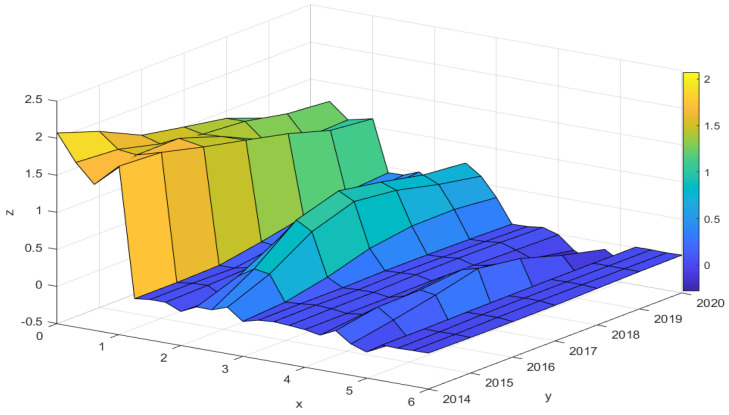
Regression results through GTWR.

**Figure 5 ijerph-19-14941-f005:**
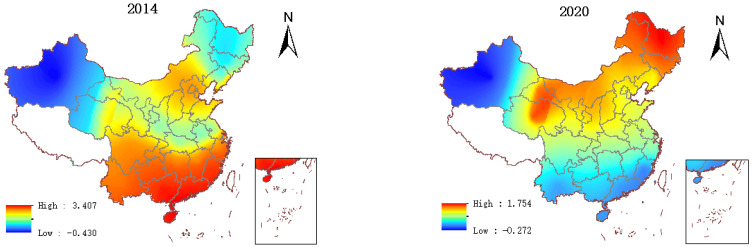
Regression coefficients for digitization in 2014 and 2020.

**Figure 6 ijerph-19-14941-f006:**
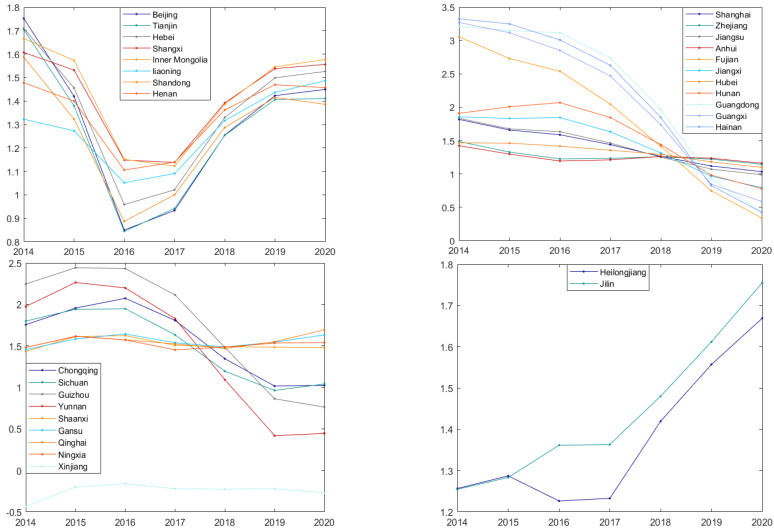
Timing fluctuations in coefficients.

**Figure 7 ijerph-19-14941-f007:**
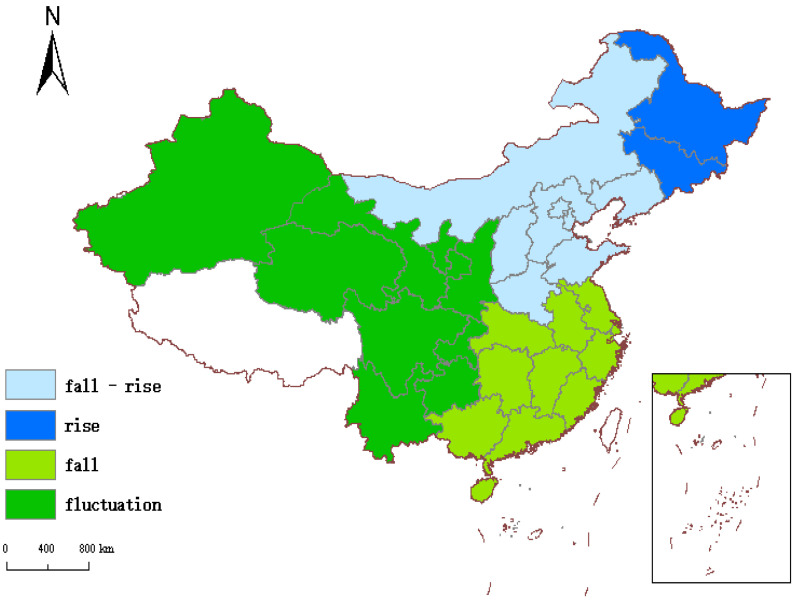
Distribution of timing fluctuations.

**Figure 8 ijerph-19-14941-f008:**
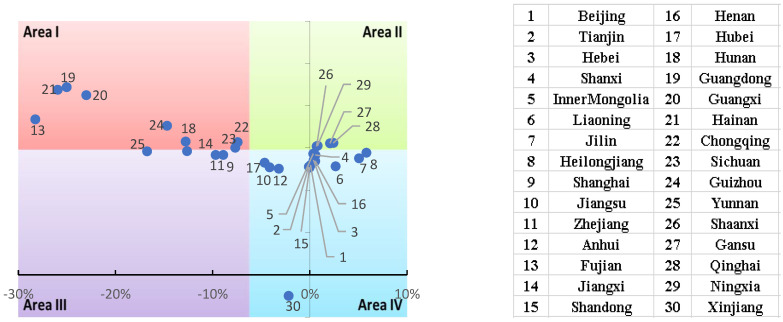
Scatter plot of the distribution of regression coefficients by province.

**Table 1 ijerph-19-14941-t001:** GTFP measurement index system.

First-Level Indicators	Second-Level Indicators	Third-Level Indicators
Input index	Capital input	Capital stock
	Labor input	Total employees
	Energy input	Energy consumption
Output index	Expected output	Real GDP
	Unexpected output	Wastewater discharge
		SO_2_ discharge
		Industrial solid wastes

**Table 2 ijerph-19-14941-t002:** Digital index system.

First-Level Index	Second-Level Index	Three-Level Index
Digital foundation	Communication access level	Telephone penetration
		Internet traffic per capita
		Optical cable line length
	Broadband access level	Mobile switchboard traffic
		Internet broadband access ports per 10,000 people
Digital input	Technology R&D investment	Technical market turnover
		R&D expenditure of industrial enterprises above scale
		Full-time equivalents of R&D personnel in industrial enterprises of above size
	Human resource level	Information transmission, software, and IT services employees as a percentage of headquarters employment
	Cultural and educational level	Per capita local financial education expenditure
Digital economy	Per capita digital economy	Total telecom services per capita
		Software business income per capita
		IT service income per capita
	Enterprise digital economy	Proportion of companies with e-commerce transaction activities
		Average business e-commerce sales
Digital applications	Enterprise digital applications	Computers per 100 people
		Websites per 100 companies
	Personal digital applications	Express business volume per capita
		Domain names per 10,000 people
		Number of pages per capita

**Table 3 ijerph-19-14941-t003:** Descriptive statistics of all variables.

Variable	Obs	Mean	Std. Dev	Min	Max
GTFP	210	1.1975	0.2660	0.8900	2.3377
DIG	210	0.1106	0.1008	0.0183	0.6410
OPEN	210	0.2324	0.2247	0.0071	1.1762
STR	210	0.5021	0.1175	0.1067	0.9750
GOV	210	0.2681	0.1126	0.1188	0.7534
GDP	210	2.7717	2.2294	0.1848	11.0761
LAB	210	8.2147	0.7344	6.3682	9.4434

**Table 4 ijerph-19-14941-t004:** Multiple collinearity test.

Variable	LAB	GDP	GOV	DIG	STR	OPEN
VIF	4.65	4.00	3.41	2.92	2.00	1.79
Mean VIF	3.13					

**Table 5 ijerph-19-14941-t005:** Regression results.

Variable	Coefficient	Variable	Coefficient
Constant	6.3770 (0.001) ***	GOV	−0.6212 (0.053) *
DIG	1.7131 (0.000) ***	GDP	0.0363 (0.048) **
OPEN	−0.3951 (0.008) ***	LAB	−0.6944 (0.004) ***
STR	0.9830 (0.000) ***		

Note: The value in parentheses is *p*: *** *p* < 0.01, ** *p* < 0.05, and * *p* < 0.1.

**Table 6 ijerph-19-14941-t006:** Test result.

Year	Z Value (*p*-Value)	Year	Z Value (*p*-Value)
2014	1.5865 (0.0710) *	2018	0.1890 (0.4020)
2015	−0.0162 (0.4960)	2019	−1.1862 (0.1050)
2016	−0.6307 (0.2810)	2020	0.6749 (0.2370)
2017	−1.4245 (0.0620) *		

Note: The value in parentheses is *p*: * *p* < 0.1.

## Data Availability

The datasets generated and analyzed during the current study are not publicly available but are available from the corresponding author on reasonable request.

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
