# Peer review of "Analysis on the Spatio-Temporal Evolution Characteristics of the Impact of China’s Digitalization Process on Green Total Factor Productivity"

_ijerph, 2022, doi:10.3390/ijerph192214941_

Round 1
Reviewer 1 Report
Dear Authors,
I’m glad I had the opportunity to revise this interesting work, I think the investigated topic is timely and it would be interesting for the readers of this Journal and the adopted method is coherent with your aim. However, some minor revisions are required before publishing it.
1. Identify the research gap (i.e., theoretical and practical relevance, research gap missing)
2. Second, considering Section 2, I suggest mobilizing part of those concepts in the other paragraphs of the literature review since I cannot revise any great value added by this section. More writings needed here to justify.
3. Third, regarding the Materials and Methods, I think that a better justification for investigating the study should be provided. What distinctive elements make its investigation interesting? What novel insights can be derived from such analysis? Then, some lines dedicated to describing the advantages of adopting the method should be added.
4. Fourth, I suggest also giving more attention to developing the discussion of your results, trying to better link it to the current scientific debate and the literature leveraged in Section 2. Reading this paragraph, I still cannot capture the distinctive elements that support the relevance of your study.
Thank you
Author Response
Dear Reviewer,
Thank you for your comments on our manuscript. These comments and suggestions have guided us to improve the content of our research. We have substantially revised our paper, as described in the attached files and manuscript.

Reviewer 2 Report
An interesting paper about GTFP of the Chinese provinces. The method of analysis (GTWR) is sound (although, a panel ARDL method like PMG might have been a better choice to examine both long-run and short-run effects) and the results are important for policy conclusions.
Comment: Table 4 presents VIF multicollineatiry test for the regressors. Although the accepted threshold for multicollinearity presence is 5 the LAB variable is very close to it (4.65). I propose authors to use also the Coefficient Variance Decomposition (CVD) test to verify their results.
Minor comment:
In abstract (line 12) the word 'regression' is missing.
Author Response

(The authors gave the same response as above.)

Reviewer 3 Report
1. The abstract did not show the main contribution and creativity of this study. The method is not clear as well. And at last, authors mentioned “Some policy suggestions are also given”, what kind of suggestion should be pointed out.
2. The research gaps are lacking in the section 2. What are critical shortcomings of current measurement of GTFP and current research about influence factors? The research gap should be proposed based on the literature review, and this research gap should guide the following contents of this study. Authors mentioned that “Most studies do not consider spatial heterogeneity and ignore the connections between geographic units, which is not in line with reality”, is that true? I have read several relevant literature. Please double check.
3. Authors have conducted abundant analyses. I suggest moving the part of policy implications after the results, also the policy implications need enhancements to exactly target the issues revealed by the analyses, for example, how to deal with the negative growth of Guangdong, Fujian and Hainan, how to take advantage of agglomeration characteristics to improve GTFP (here are just two examples, there are still other points that authors should pay attention).
4. The part of conclusion should be placed at the end, and contain the main creativity, finding (not too much), contribution, limitations of this study, and future research.
Author Response

(The authors gave the same response as above.)

Reviewer 4 Report
Overall a very interesting study which deserves attention and promotion. Even though digitalisation is a current debated topic in many studies, the authors have managed to provide a compelling work with the particularities of this subject in China. The quantitative part of the study is very strong, the conclusions' part needs an improvement - to mention which are the limitations of this study? Otherwise congratulations to the authors
Author Response

(The authors gave the same response as above.)
